# The Role of Multiply-Fortified Table Salt and Bouillon in Food Systems Transformation

**DOI:** 10.3390/nu14050989

**Published:** 2022-02-26

**Authors:** Dipika Matthias, Christine M. McDonald, Nicholas Archer, Reina Engle-Stone

**Affiliations:** 1Bill & Melinda Gates Foundation, Seattle, WA 98109, USA; 2Departments of Pediatrics, and Epidemiology and Biostatistics, University of California, San Francisco, CA 94143, USA; christine.mcdonald@ucsf.edu; 3Department of Nutrition and Institute for Global Nutrition, University of California, Davis, CA 95616, USA; renglestone@ucdavis.edu; 4CSIRO Health and Biosecurity, North Ryde, Sydney 2113, Australia; nicholas.archer@csiro.au

**Keywords:** salt, bouillon, condiment fortification, fortified foods, micronutrient deficiencies, sodium, large scale food fortification, fortification innovations, food systems, non-communicable diseases

## Abstract

Our global food system lacks the critically needed micronutrients to meet the daily requirements of the most at-risk populations. Diets also continue to shift toward unhealthy foods, including the increased intake of salt. While most countries exceed the WHO’s recommended levels, sodium does play an essential physiological role. Table salt and other salt-containing condiments, such as bouillon, also have cultural importance, as they are used to enhance the flavor of foods cooked at home. Given their universal consumption across income classes and both urban and rural populations, these condiments are an integral part of the food system and should, therefore, be part of its transformation. Fortification of salt and salt-containing condiments can play a catalytic role in the delivery of population-wide nutritional and health benefits. With relatively consistent levels of intake across the population, these condiments hold high potential for delivering micronutrients beyond iodine while also reducing concerns related to high micronutrient intake, particularly so in countries where the industries are relatively consolidated. As a flexible and complementary strategy to an evolving food system, fortification levels can also be adjusted over time to ensure micronutrient delivery targets continue to be achieved as the system improves, whether through lower intakes of sodium in line with WHO recommendations, enhanced consumption of nutrient-dense foods, and/or broader adoption of biofortified crops. Future areas of innovation are required to realize this vision, including developing affordable salt substitutes to meet cost requirements of consumers in low-and middle-income countries, improving the stability and bioavailability of the micronutrients in condiments so that delivery targets can be reached without affecting sensory attributes, and the development of efficient systems for monitoring population intake and micronutrient status to inform fortification program design and management. Rather than being considered antithetical to the transformation, multiply-fortified salt and bouillon can strengthen our ability to meet the cultural, sensory, nutritional, and health needs of an evolving food system.

## 1. Introduction

Over 2 billion people are at risk for micronutrient deficiencies globally [1], a figure that has remained unchanged for decades and has only been further exacerbated by the global COVID-19 pandemic that has reduced access to micronutrient-rich diets in low-income countries due to disrupted supply chains and diminished incomes, with a disproportionate impact on women and young children whose higher micronutrient needs place them at higher risk for deficiency [2]. Even prior to the COVID-19 pandemic, micronutrient-rich diets, including fruits, vegetables, and animal protein, have been out of reach for over 1.5 billion people due to a cost that exceeds household income [3]. The Green Revolution that began in the 1960s dramatically enhanced the agricultural productivity of wheat and rice, reducing hunger and poverty for hundreds of millions of consumers globally. Yet, it may have also had the unintended consequence of exacerbating micronutrient malnutrition, due in part to a corresponding decrease in the production of micronutrient-dense crops, such as legumes, as well as to the milling and polishing of cereal staples that remove between 20% and 95% of their already low levels of micronutrients [4]. Analysis of national food supply data indicates that micronutrient availability is inadequate in several low- and middle-income countries (LMIC) and that the micronutrient density of the food supply decreased in sub-Saharan Africa from 1961 to 2011 [5]. Further, in South Asia, time trend data reveal an increase in anemia among women of reproductive age with increasing production of wheat and rice and lower overall iron density in the diet [4]. Milling, as well as cooking practices, such as boiling rice in excess water, further reduce the micronutrient content at the point of consumption, particularly for a range of B-vitamins, zinc, and iron [6,7]. From farm to fork, food systems are not optimized for the delivery of micronutrients to those most in need.

The health and economic consequences of a food system with low micronutrient density are significant to both individuals and entire nations. Multiple micronutrients play a critical role in immune system function and, when deficient, increase susceptibility to disease [8]. Vitamin A and zinc deficiencies account for a respective 157,000 and 116,000 child deaths across Africa, Asia, and Latin America each year [9], while dietary iron deficiency is the seventh leading cause of DALYs (disease-adjusted life years) for children 0–9 years of age [10]. Micronutrients are critically important to a healthy pregnancy. Deficiency in micronutrients such as iron, iodine, zinc, folate, and vitamin B12 can increase the risk of several adverse pregnancy outcomes, including low birth weight, preterm birth, and neural tube defects. In 17 of the 18 countries with nationally representative data available, more than 20% of women of reproductive age are deficient in zinc, a micronutrient that is critical for the optimal growth and development of children [11]. Finally, micronutrient deficiency, particularly of iron and iodine, has been implicated in GDP (gross domestic product) losses of between 0.8% and 2% due to their critical role in cognitive development and work productivity [12].

Increasing the micronutrient density of the food system is a complex endeavor, requiring a range of both demand- and supply-side interventions. On the supply side, biofortified crops, fortifying staple foods at the point of food processing, and improving both access and affordability of micronutrient-dense diets have been core intervention strategies, while demand-side interventions focus on raising consumer awareness of nutrient-dense foods. While no single intervention will transform the system, food fortification of basic staple cereals and condiments was ranked by the Copenhagen Consensus in 2008 as one of the most cost-effective interventions in health and development [13]. In particular, the fortification of salt with iodine, at only $0.05 per person per year that generates $30 in economic return for every $1 invested, remains a cornerstone of food fortification programs [13]. The objective of this paper is to describe the rationale for expanding the range of micronutrients carried by salt and salt-containing vehicles, such as bouillon (i.e., multiply-fortified salt and bouillon). Fortification of table salt and bouillon with multiple micronutrients holds enormous potential for delivering additional population-wide nutritional and health benefits and is indeed compatible with sodium reduction initiatives to reduce population risks of non-communicable diseases.

## 2. Current and Emerging Dietary Trends

Global dietary patterns have undergone drastic changes in the past three decades as a result of increased urbanization, greater participation of women in the formal workforce, increased consumption of food away from home, and other factors [14]. The Global Dietary Database Consortium’s (GDDC) systematic analysis of consumption data from 187 countries revealed that although global consumption of healthier foods and nutrients (the authors classified the following items as “healthy”: fruits, vegetables, beans and legumes, nuts and seeds, whole grains, milk, total polyunsaturated fatty acids, fish, plant omega-3s, and dietary fiber) increased modestly between 1990 and 2010, consumption of unhealthy foods and nutrients (the authors classified the following items as “unhealthy”: unprocessed red meats, processed meats, sugar-sweetened beverages, saturated fat, trans fat, dietary cholesterol, and sodium) increased to a greater extent during the same time period [15,16]. The analysis also revealed striking geographic and economic disparities. Improvements in the consumption of healthy foods and reductions in the consumption of unhealthy foods were greatest in high-income countries, whereas both dietary patterns worsened in many low-income countries [15]. Notably, middle-income countries exhibited the largest improvement in the consumption of healthy foods but also the largest deterioration in consumption of unhealthy foods [5]. Similar economic and geographic differences in food availability and consumption are also evident within countries. For example, a recent analysis of foods available within 44 retail outlets across four states in India revealed increased availability (both in terms of absolute number and relative proportion) of packaged food in urban vs. rural areas [17]. Unsurprisingly, packaged foods were 50% less healthy and over three times as expensive as unpackaged food. An understanding of these geographic and socioeconomic differences in food availability and consumption is necessary to inform the design and targeting of effective policies and interventions that will achieve the intended public health outcomes among the most relevant segments of the population.

At the global level, there is increasing attention to hypertensive disorders and the role of dietary sodium. Although sodium is an essential nutrient that plays a critical role in many life-sustaining processes [18], excessive sodium intake is associated with an increased risk of hypertension and various non-communicable diseases (NCDs) [19,20]. Interestingly, in 2010, only six of the 187 countries examined by the GDDC met the WHO’s recommendation for sodium intake of <2 g/day (approximately equivalent to <5 g salt per day) [16]. Additionally, more detailed analyses revealed that global mean sodium intake in 2010 was 3.95 g/day (10.06 g salt per day) and had not changed significantly since 1990 [21]. Asian regions tended to have the highest salt intakes, whereas countries in sub-Saharan Africa, Latin America and the Caribbean, and Oceania tended to have lower estimated intakes in relative terms, although there was more uncertainty in these lower estimates. Globally, salt intake increased from 4 g to 5.6 g per capita per day from 2010 to 2017 [22].

## 3. The Role of Salt and Bouillon in Food Systems Transformation

Improving the nutritional quality of the food supply through increased micronutrient density of foods was identified as a candidate double-duty action to address multiple forms of malnutrition [23]. Food fortification is, therefore, an integral part of food systems change and can be considered a double-duty intervention as long as the program is designed to avoid the excessive consumption of micronutrients or of the fortification vehicles themselves. Multiply-fortified salt and bouillon are especially well suited to the need, given their primary role as flavor enhancers for foods that are cooked at home. Building on a successful history of salt iodization, both salt and bouillon are increasingly considered fortification vehicles for a range of other micronutrients [24,25,26]. These condiments are uniquely suited to meet the criteria set out by the WHO in the 2006 food fortification guidelines [27]: they tend to be centrally processed, are consumed by a large proportion of the population in consistent quantities, and can be fortified with available micronutrient fortificants (though additional research is underway to address issues such as micronutrient stability [28] and bioavailability [29,30]). Salt and bouillon (where commonly used) are universally consumed across all income classes [31,32,33]. A survey of fortifiable food use in four African countries found that salt was most commonly available in households; samples of fortifiable (industrially processed) salt were collected from 72–86% of households compared with <32% for wheat and maize flour [32]. In a second analysis from West Africa, bouillon was consumed by 79–99% of women in the past week, and the average frequency of consumption was 5–7 days per week [33]. While some differences in bouillon consumption were observed between urban and rural strata in Burkina Faso and Niger, the reach of bouillon remained high (>~50%) in all strata, and in other countries, there were no differences in reach by residence (98% vs. 100% in Senegal; 93% vs. 97% in Cameroon) [33].

Modeling studies have demonstrated the potential for the fortification of salt and bouillon to fill dietary micronutrient gaps. In some contexts, these condiments may be consumed within a narrower range of intakes compared to other potentially fortifiable foods, which facilitates the selection of fortification levels that balance micronutrient inadequacy and excess. Modeling of national dietary intake data (collected by 24-h dietary recall) from Cameroon showed that the distribution of bouillon intake was less skewed than the distribution of wheat flour intake and that bouillon fortification with vitamin A and folic acid could achieve similar reductions in inadequate intake, with a lower likelihood of exceeding the tolerable upper intake level (UL) compared to wheat flour fortification [34,35]. Figure 1 provides a visual depiction of these benefits, with almost zero percent of the population at risk of exceeding the UL for both vitamin A and folic acid at the modeled fortification levels for bouillon. In yet another modeling study that utilized national household survey data from Cameroon, Ghana, and Haiti, bouillon fortification with micronutrients was predicted to reduce the prevalence of inadequate intakes among women, men, and preschool children by 15–33 percentage points (pp) for vitamin A (120 µg retinol/g), 11–33 pp for folate (80 µg folic acid/g), 12–67 pp for vitamin B12 (1.2 µg/g), 5–12 pp for iron (5 mg/g, assuming 2% absorption), and 14–42 pp for zinc (3 mg/g), suggesting that the reach and amount of bouillon consumed in these populations is sufficient to achieve a public health impact across multiple micronutrients [36]. Likewise, modeling of detailed dietary intake data has also been used to inform fortification levels for research on multiply-fortified salt to select levels that best balance efficacy and safety (CM, personal communication).

Salt and bouillon, therefore, hold high potential for reducing the health burden from micronutrient deficiencies, particularly in countries where the industries are moderately consolidated. Iodized salt has already demonstrated this potential at scale, with a significant impact on goiter reduction and improved IQ since its introduction almost a century ago [37,38]. A growing evidence base for the dual fortification of salt with iodine and iron indicates positive impacts on hemoglobin, iron status, anemia, and iron deficiency anemia [39]. While a variety of interventions exist to deliver specific nutrients (including the delivery of supplements or promotion of specific nutrient-dense foods), large-scale food fortification tends to be cost-effective relative to other micronutrient-specific interventions [40] and relative to other programs to improve maternal and child nutrition and health [41].

Given the ubiquitous consumption of salt and bouillon and the opportunity for enormous health impacts, fortifying these vehicles with multiple micronutrients can dramatically change their role within the food system. Rather than being viewed as bad actors, they can be used as essential vehicles for bridging gaps in the dietary intake of critical micronutrients. Concerted efforts to improve nutritional status and related health outcomes through basic elements of the diet are indeed an important aspect of the food system transformation.

## 4. Salt and Bouillon Fortification within the Global Effort to Reduce Dietary Sodium Intake

Salt and bouillon contain high sodium levels of approximately 40% and 20–30%, respectively. As outlined above, global consumption of sodium is above recommended levels, and the average daily intake is continuing to increase [21,22,42]. Ironically, countries that are well suited for multiply-fortified salt and bouillon to address gaps in micronutrient intake are also the regions with the highest prevalence of hypertension, in particular LMICs in sub-Saharan Africa and South Asia [43,44,45]. Further compounding the high incidence is the lower awareness of hypertension and treatment resulting in greater cardiovascular disease (CVD) severity and secondary complications in these LMIC regions compared with high- and middle-income countries [43,46].

Sodium reduction has been identified as a cost-effective mechanism to reduce hypertension, CVD, and NCD mortality across all economic levels, including in LMICs [47]. Ideally, sodium reduction strategies should be developed in accordance with distinct dietary patterns. The dietary source of sodium can vary drastically according to geography and income. There are three main sources of sodium in the diet: (1) sodium naturally present in food, (2) sodium added to packaged or processed foods, and (3) sodium added during cooking or prior to eating at the table (discretionary/table salt). In high-income countries, the vast majority of sodium is consumed from processed foods; however, in lower-income countries, most sodium comes from salt (or seasoning such as bouillon) that is added during cooking (i.e., discretionary salt) [48]. Thus, attempts to reduce sodium levels in discretionary salt and bouillon consumed in LMICs hold strong potential for reducing sodium intake and, in turn, making progress toward the WHO’s 2030 goal of reducing premature deaths from NCDs by 25% [49].

A recent analysis of discretionary salt intake in 33 studies revealed an inverse correlation between a country’s GDP per capita and the proportion of daily salt intake from discretionary salt. For every $10,000 GDP per capita, the proportion of daily salt obtained from discretionary sources was lower by 8.7% [42]. Therefore, the amount and source of salt consumed by different segments of the population, as well as the various purchasing and/or procurement modalities, have important implications for the design of sodium reduction and fortification efforts alike.

While LMICs have the highest rates of hypertension and mortality due to CVD compared with high- and middle-income countries, there are fewer salt reduction initiatives in these geographies [50]. Interventions for population-wide sodium reduction include consumer education to reduce intake of high salt foods (e.g., media campaigns or front of pack labeling), reformulation of foods to contain lower sodium content [48], and government-led initiatives to set voluntary targets or mandatory maximum levels [47,50,51]. Options for reducing sodium intake from discretionary use of salt and bouillon in home cooking include consumer education to reduce the addition of salt to foods while cooking in the home environment [52] and increasing the use of salt substitutes [53].

With the global effort to reduce salt intake to tackle rising NCD incidence, there is concern that this trend is incompatible with the utilization of salt as a fortification vehicle. Reducing salt intake could: (i) lower the effectiveness of salt fortification programs, resulting in an increased incidence of micronutrient deficiency, particularly iodine at present [54,55] and/or (ii) reduce the effectiveness of salt and salt-containing condiments for additional micronutrients beyond iodine [56]. Additionally, fortified salt or bouillon may be seen to have greater health benefits by a consumer (e.g., positive consumer bias toward foods with front of pack nutrient claims [57]) encouraging increased consumption and a higher intake of salt [58]; however, there is no clear evidence to support the concern that fortification leads to higher consumption of the vehicle.

The public health goals of reducing salt intake while fortifying salt to increase micronutrient intake (e.g., through salt iodization) are indeed compatible, given that the concentration of micronutrients can be adjusted as the intake of salt/bouillon is reduced (further referred to in the paper as titration) [54,59]. Success will require continuous monitoring of salt intake (e.g., through 24-h urinary sodium measures and dietary intake surveys) and the level of micronutrient intake (e.g., comprehensive dietary assessments to understand the contribution of iodine from fortified or other dietary sources) to enable titration of the fortificants in salt to the level required (i.e., to reduce the prevalence of inadequate micronutrient intake to the greatest extent possible while minimizing the proportion of the population with intake above the tolerable upper intake level) [55,60]. While there appears to be significant consensus on the compatibility of micronutrient fortification and sodium reduction initiatives [54,58,59,61,62], there are few reported examples in practice, likely due to few countries successfully reducing sodium intake [22]. One reported example is Switzerland, where voluntary iodine levels in salt have been increased in response to successful salt reduction to ensure longer-term adequate intake of iodine [63].

While policies for fortification and reduction of salt intake are compatible, the impact of salt reduction would be specific to the context and country [59]. Key factors affecting strategies to titrate fortification levels over time would include the amount and source of salt reduction across the population (i.e., sodium reduction in the household during cooking vs. from processed foods) and changes in the dietary pattern of the population, including through improved nutritional quality of the diet or availability of new fortified foods. Further, given the close synergy between these two public health strategies, it is important that there is (i) better coordination at the country level across both fortification and sodium reduction programs/policies and (ii) increased communication and messages developed to reinforce their complementarity. For example, in Italy, a previous communication strategy was “Little salt, but all iodized” [59]. Such communication strategies may overcome concerns about the increased consumption of salt due to the perceived health benefits from fortification. These campaigns should ideally communicate both the positive health effects of iodized salt and the negative health effects of excessive salt intake to support the behavior change. In fact, another study in Italy found significant improvements in children’s intake of iodine through salt but little change in the intake of high-sodium processed foods after an educational campaign that addressed both [64]. The authors hypothesized that this might be because the children were educated on the benefits of iodine, but the campaign focused less on the link between excess sodium intake and cardiovascular disease. Efforts targeting the availability and accessibility of processed foods high in sodium vs. lower-sodium alternatives may also be needed.

## 5. Areas of Further Research Needed

Several areas of innovation are required to meet the dual goal of broadening the micronutrients added to salt and bouillon while also meeting salt reduction targets in LMICs. These include the development of low-cost sodium substitutes, micronutrient innovations to ensure sensory, cost, and stability attributes can be met, and dietary surveillance to support the continued optimization of food fortification programs. These types of innovations are critical for maximizing the potential role of salt and bouillon within the food systems transformation effort. Low-cost sodium substitutes would allow governments to leverage a broader range of salt-containing condiments as vehicles of essential micronutrients. Micronutrient innovations would improve the technical and economic feasibility of doing so, and new surveillance tools would enable critical adjustments to the range and levels of micronutrients needed in the population as dietary patterns and intervention programs evolve.

### 5.1. Sodium Substitutes

A key challenge for sodium reduction is the loss of taste when salt is removed, combined with the fact there is currently no direct salt substitute available. Partial substitution with other non-sodium salts is an option that is widely used in many foods (e.g., potassium chloride or ammonia); however, these have negative sensory effects when used at higher concentrations. Due to the selective nature of the ion channel(s) associated with the detection of salt taste, there may not be a true substitute that can replace salt [48]. Optimal methods for sodium reduction in current/future micronutrient vehicles containing high salt levels such as bouillon and other condiments require further research. Thus, salt reduction of high-sodium-containing foods such as condiments and bouillon will likely involve multiple strategies for reformulation to enable significant reduction [48]. In the case of bouillon, additional options include reformulating to exploit cross-modal interactions (taste-taste or taste-aroma) to enhance saltiness and overall flavor while reducing the salt content. In such a way, there is the potential to maintain overall flavor intensity and consumer satisfaction while enabling a reduction in salt levels.

Salt also delivers other techno-functional properties beyond making food taste appetitive. For example, salt contributes to the size and compression of a bouillon cube, and salt reduction will require a filler to maintain the cube size and compression. Further, salt is a low-cost ingredient, and reformulation to reduce salt and/or replacement with other ingredients to account for the loss of flavor or other techno-functional properties will have added costs above the cost saving from the salt removed. This increased cost will likely be passed onto the consumer, which could impact its affordability and thereby reduce the frequency and amount consumed across a population. While it is possible to titrate fortification levels higher if consumption decreases, it will be important to ensure cost increases due to sodium reduction and fortification do not reduce consumption by the lowest socioeconomic groups that are likely to gain the greatest benefit from fortified foods/programs.

### 5.2. Micronutrient Innovation

Expanding the micronutrients in salt and bouillon beyond iodine holds great potential for reducing a range of deficiencies in a single product, yet such products can introduce both stability and sensory challenges. Iron alone in various double-fortified salt (DFS) formulations, in which an iron-containing premix is added to salt, have revealed an unacceptable sensory profile, including dark specks that are particularly acute in poor-quality premix (in which the encapsulation does not fully cover the dark iron) or when the premix is added to poor-quality salt (in which the moisture can erode the encapsulation) [65]. Further, when the iron encapsulation dissolves with heat during cooking, it darkens the food and can affect consumer acceptance [66]. Iron can also destabilize iodine [67]. Research efforts are currently underway to address these challenges using a whiter and less reactive form of iron, such as ferric pyrophosphate, with added enhancers to compensate for its lower bioavailability [30,68]. While a promising iron compound for the fortification of double and multiply-fortified salt, competitive approaches could further optimize the bioavailability, sensory impact, and combinability of iron with other micronutrients in salt and salt-containing vehicles.

Vitamin A stability is also a challenge, particularly when stored for long periods of time near other micronutrients and then exposed to high temperatures, such as in multiply-fortified bouillon. Commercially available forms of encapsulated vitamin A have limited stability in such conditions, requiring significant overages to ensure that target delivery levels are available at the time of consumption. Several research efforts are underway to strengthen vitamin A encapsulation with both natural and synthetic polymers [69]. These polymers are pH-sensitive, protecting vitamin A through the cooking process and releasing at low pH in the stomach to enable absorption in the proximal duodenum. However, these technologies have yet to be commercialized.

Finally, in some geographies, such as Ethiopia, where neural tube defects are of great public health concern, variants of double-fortified salt with iodine and folic acid are now in development. When added to the iodine overspray, folic acid can turn the salt a pale yellow. Consumer acceptability still needs to be assessed. If unacceptable, new methods of integrating folic acid into salt, such as the development of a concentrated folic acid premix that is then blended into salt, will need to be developed.

### 5.3. Dietary Surveillance

To ensure national fortification programs are operating as efficiently and effectively as possible, policies should be reviewed and modified on a regular basis to account for possible changes in the prevalence of micronutrient deficiency, consumption of the vehicle, and/or other dietary sources of micronutrients. This process requires up-to-date information on dietary intake, coverage and quality of the fortified vehicle, and the status of other micronutrient interventions. Unfortunately, many of the current data collection methods that provide high-quality information on individual food and nutrient intake (such as observed, weighed food records) are expensive, time-consuming, and labor-intensive. Innovations to develop and apply assessment tools that are more field-friendly may increase availability of the information needed to adjust fortification programs. For example, tablet-based programs have been developed to facilitate the collection of dietary intake data [70], and new statistical tools can decrease the time and cost of data analysis [71]. Where there are questions related to the coverage and contribution of other fortification programs, targeted assessments such as the Fortification Assessment Coverage Tool may suffice [72]. In addition, point-of-care biomarker tests merit exploration as a source of information on changes in population micronutrient status. Other possibilities include exploring ways in which secondary data sources can be effectively utilized, such as information collected at the retail level, or through national household surveys, such as Household Income and Expenditures Surveys, that collect information on household food acquisition or consumption [73]. While efforts to address micronutrient deficiencies and chronic disease are often managed by separate organizations, collaboration between these groups may facilitate the joint collection of information that can inform multiple policy objectives; for example, information on salt consumption and/or urinary sodium excretion could be used to inform programs to fortify salt as well as sodium reduction strategies. In addition to efforts to streamline the collection and analysis of these data, clear pathways must be in place for making this information available in relevant policy discussions.

## 6. Summary

Our global food system lacks the critically needed micronutrients to meet the daily requirements of the most at-risk populations. Due to the phenomenal success of salt iodization over the last 30 years, we would be remiss not to consider expanding the range of micronutrients in salt as a means of supporting the transformation of the food system. The factors that have made salt iodization so successful, including its universal consumption and narrow range of consumption across the population, are even more relevant today, as standards authorities grow increasingly concerned with the risks of excessive intake and potential toxicity as multiple micronutrient interventions are deployed. To maximize the effectiveness of fortification programs, the policy focus for adding micronutrients should be on discretionary salt rather than on salt contained within processed foods, as sodium intake comes largely from the salt used during cooking and at the table for those populations with the highest gaps in intake.

On the surface, the strategy to fortify salt with multiple micronutrients might seem incompatible with guidelines from global and local health authorities to reduce salt intake to reduce the risk of hypertension and other NCDs. While there is concern that micronutrients can lead to the increased intake of salt, in truth, there is no evidence that iodization has increased levels of salt consumption. Additionally, fortification is a flexible strategy, as the concentration of micronutrients in salt can be adjusted over time to meet population intake targets as salt consumption levels decline in response to sodium reduction initiatives, as has been demonstrated in Switzerland. However, there are several areas of research and innovation that could better support the deployment of these twin strategies, including the development of cost-effective salt substitutes and micronutrient innovation to ensure that sensory, bioavailability, and stability targets can be maintained as micronutrients beyond iodine are added to salt, potentially at increasingly higher concentrations over time as salt reduction programs take effect. Finally, new methods for cost-effectively capturing salt intake in the population are required for understanding changing population intakes and maximizing the impact of a combined salt fortification and NCD reduction strategy. While fortification of salt and salt-containing condiments is not a panacea, it holds enormous potential for being a critical anchor for meeting the dietary requirements across a broad range of micronutrients for those with the greatest need, as we collectively move toward a food system that delivers better health for all.

## Figures and Tables

**Figure 1 nutrients-14-00989-f001:**
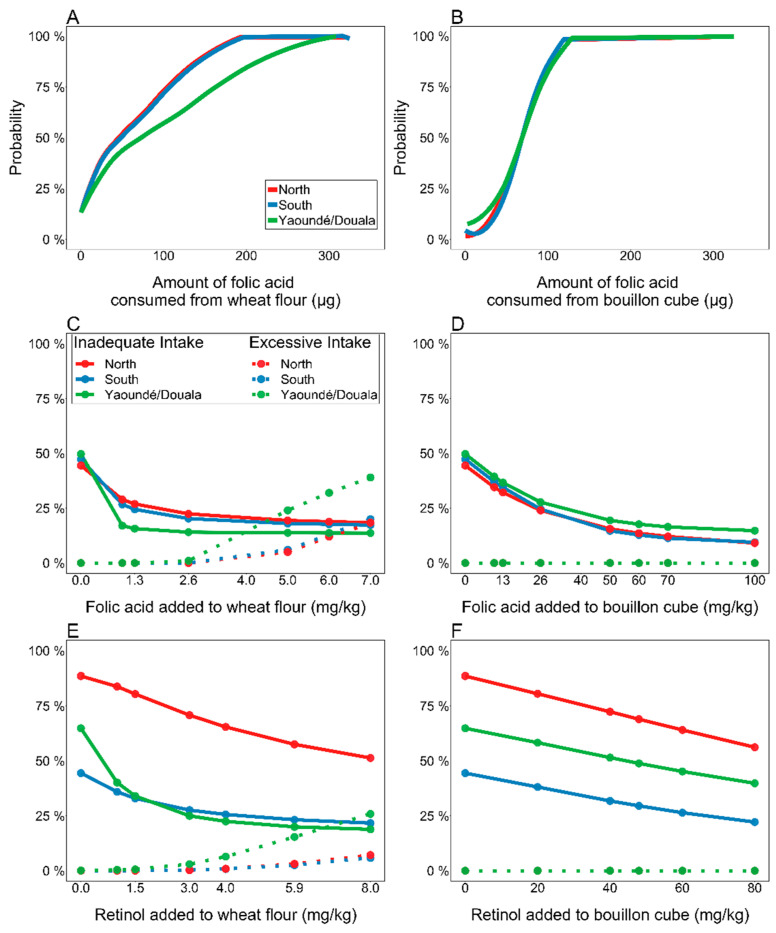
Distribution of folic acid consumption from fortified wheat flour or bouillon cube intake at equivalent fortification levels based on mean national intake, and relative effects of fortification of each vehicle on inadequate and excessive intakes among Cameroonian children 1–4 years of age. For Panels C–F, solid lines indicate prevalence of inadequate intakes, and dotted lines indicate prevalence of intakes above the UL. (**A**) Cumulative probability distribution of folic acid intake from wheat flour, at 2.6 mg/kg (designed to provide 70 μg/day based on average national intake of 27.0 g/day). (**B**) Cumulative probability distribution of folic acid intake from bouillon cube, at 70 mg/kg (designed to provide 70 μg/day based on average national intake of 1.0 g/day). (**C**) Prevalence of inadequate and excessive folate intakes at different levels of fortification of wheat flour. (**D**) Prevalence of inadequate and excessive folate intakes at different levels of fortification of bouillon cube. (**E**) Prevalence of inadequate and excessive vitamin A intakes at different levels of fortification of wheat flour. (**F**) Prevalence of inadequate and excessive vitamin A intakes at different levels of fortification of bouillon cube. This figure is reproduced from Engle-Stone et al. [34] under a Creative Commons attribution license.

## Data Availability

Not applicable.

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
