# Peer review of "The Role of Multiply-Fortified Table Salt and Bouillon in Food Systems Transformation"

_nutrients, 2022, doi:10.3390/nu14050989_

Round 1

Reviewer 1 Report

The Article of Dipika and co-Authors analyse the role of salt and bouillon as essential sources and crucial tools in providing a population of fundamental micronutrients (iodine vitamin A iron and so on), particularly fragile subjects as childbearing females, children and those living in LMIC. Moreover, the paper analyses possible strategies to maintain an adequate intake of fortified salt and bouillon together with the efforts to reduce sodium consumption particularly in LMIC regions. Finally, the Authors consider possible areas of further research innovation considering sodium substitutes, salt fortified with different micronutrients and novel smart methodologies to evaluate daily salt intake.

The paper is particularly sound, clearly written and easy to follow and I have only few remarks.

Page 2 line 59 “DALYs” acronym for? Line 66 “GDP” acronym for? Please state.

Please include also iodine among micronutrients critically important to a healthy pregnancy (Page 2, line 60)

Page5 lines 232-244, among strategies aimed to improve the awareness of importance of iodized salt use  and to reduce overall consumption of sodium-enriched foods please also cite and comment another Italian study that fits very well with these two goals (Watutantrige-Fernando S et al Efficacy of educational intervention to improve awareness of the importance of iodine, use of iodized salt, and dietary iodine intake in northeastern Italian schoolchildren. Nutrition. 2018)

Author Response

1. Page 2 line 59 “DALYs” acronym for? Line 66 “GDP” acronym for? Please state.

Response: Have added explanations for both acronyms. DALYs stands for Disease Adjusted Life Years. GDP standards for Gross Domestic Product.

2. Please include also iodine among micronutrients critically important to a healthy pregnancy (Page 2, line 60)

Response: Yes, thank you for noting this important omission.  We have added “iodine” to the sentence as suggested.    

3. Page 5 lines 232-244, among strategies aimed to improve the awareness of importance of iodized salt use  and to reduce overall consumption of sodium-enriched foods please also cite and comment another Italian study that fits very well with these two goals (Watutantrige-Fernando S et al Efficacy of educational intervention to improve awareness of the importance of iodine, use of iodized salt, and dietary iodine intake in northeastern Italian schoolchildren. Nutrition. 2018)

Response: Thank you for sharing this reference.  We have read this study and have included the following statement in lines 286-294 of the paper:

These campaigns should ideally communicate both the positive health effects of iodized salt and negative health effects of excessive salt intake to support the behavior change. In fact, another study in Italy found significant improvements in the children’s intake of iodine through salt, but little change in the intake of high sodium processed foods after an educational campaign that addressed both. The authors hypothesized that this may be because the children were educated on the benefits of iodine, but the campaign focused less on the link between excess sodium intake and cardiovascular disease. Efforts targeting the accessibility and price of high-sodium processed foods vs lower-sodium alternatives may also be needed.

Reviewer 2 Report

Overall comments:

  1. Lines 1 and 31 and elsewhere. What does “multiply fortified table salt” mean?  Does it refer to salt fortified with multiple nutrients?  Suggest you use another term.    
  2. What is the objective of this manuscript? Add a paragraph at the end of the introduction that clearly states it for the reader. 
  3. There are no figures or tables in the manuscript. Can any of the facts presented be transformed into visuals to help the reader understand main concepts?
  4. For section 3, I’m expecting a paragraph that more directly ties salt and bouillon fortification with food systems transformation. Same for sections 4 and 5—what is the direct relationship with food systems transformation?
  5. Section 6. If salt is already fortified and has been for decades, how does more fortified salt transform the food system?  Overall, make the link more explicit between fortified salt (and bouillon) and food system transformation. 

Detailed comments:

  1. L49, iron-deficiency anemia. Are the data in reference [4] based on hemoglobin alone or also a measure of iron deficiency.  The author of reference [4] is in the agricultural sector and may not know the importance of distinguishing anemia from iron-deficiency anemia.
  2. L51-2. Discarding excess cooking water would reduce water-soluble vitamins from unfortified rice.  Are there important vitamins, from a public health perspective, that should be highlighted in this sentence or paragraph?
  3. L80-2. Add a reference(s) for these concepts. 
  4. How was “healthy” defined in this study?  The same as in studies 13 and 14?
  5. Please describe what the intended outcome is. 
  6. Sections 2 and 3. The font size (type?) is different between these two sections.  Make uniform across the document. 
  7. L120-3. The second part of the sentence refers to fortification, but the first part does not.  Is this a generic statement that is not specific to fortification?
  8. Are availability of and access to health services a further factor?
  9. Not clear what this means:  “have higher health benefit by a consumer encouraging increased consumption”. 
  10. Add reference(s) to support the second part of the sentence.
  11. L223, 234 and elsewhere. Does the use of titration in this sentence imply laboratory measurement of iodine in salt?
  12. Remove “which”. 
  13. L289-90. The first part of the sentence is missing a word. 
  14. L298-9. Lower case vitamin.
  15. Any household surveys or specifically income and expenditure surveys?

Author Response

1. Lines 1 and 31 and elsewhere. What does “multiply fortified table salt” mean?  Does it refer to salt fortified with multiple nutrients?  Suggest you use another term.    

Response: Thank you for this comment.  The term “multiply fortified table salt” refers to the fortification of table salt with multiple micronutrients.  We have added a hyphen between “multiply” and “fortified” (i.e. multiply-fortified) in the title and have clarified the term in lines 82-83 of the text. 

2. What is the objective of this manuscript? Add a paragraph at the end of the introduction that clearly states it for the reader. 

Response: The main goal of the paper is to highlight the promise of table salt and bouillon to deliver critically needed micronutrients to at risk populations, while also being compatible with salt reduction strategies, as a dual approach improving the food system through salt. The last section of the introduction has been updated to state the objective of the review (line 79):

The objective of this paper is to describe the rationale for expanding the range of micronutrients carried by salt and salt-containing vehicles, such as bouillon (i.e., multiply-fortified salt and bouillon). Fortification of table salt and bouillon with multiple micronutrients holds enormous potential for delivering additional population-wide nutritional and health benefits and is indeed compatible with sodium reduction initiatives to reduce population risks of non-communicable diseases.

3. There are no figures or tables in the manuscript. Can any of the facts presented be transformed into visuals to help the reader understand main concepts?

Response: We agree with this comment and have asked the Nutrients editors about whether it would be acceptable to use a graphic from a previously published paper, which we have referenced on line 166.  We believe this graphic visually and effectively supports our claim that salt-containing condiments, such as bouillon, can significantly improve population intakes of multiple micronutrients, without the risk of exceeding upper limits as compared to other vehicles such as wheat. This is largely due to the narrow range of bouillon consumption across the population between those with lowest and highest intakes.  

4. For section 3, I’m expecting a paragraph that more directly ties salt and bouillon fortification with food systems transformation. Same for sections 4 and 5—what is the direct relationship with food systems transformation?

Response: Thank you for this suggestion.  We have added the following text to Section 3 (lines 215-219) to further clarify why and how multiply-fortified salt and bouillon are an important part of the food systems transformation effort:

Given the ubiquitous consumption of salt and bouillon and the opportunity for enormous health impacts, fortifying these vehicles with multiple micronutrients can dramatically change their role within the food system. Rather than being viewed as bad actors; they can be used as essential vehicles for bridging gaps in dietary intake of critical micronutrients. Concerted efforts to improve nutritional status and related health outcomes through basic elements of the diet are indeed an important aspect of the food system transformation.

Section 4 highlights that this can be done alongside efforts to reduce the overall intake of sodium. Efforts to reduce sodium levels in food (i.e. through food reformulation or consumer education) are also a contribution to the transformation of the food system; therefore, we have not further highlighted the food systems transformation link here.  However, we have added a few sentences to the end of the first paragraph of Section 5  (lines 312-318) to highlight the catalytic role that these innovations can play in effecting the change:

These types of innovations are critical for maximizing the potential role of salt and bouillon within the food systems transformation effort. Low-cost sodium substitutes would allow governments to leverage a broader range of salt-containing condiments as vehicles of essential micronutrients.  Micronutrient innovations would improve the technical and economic feasibility of doing so, and new surveillance tools would enable critical adjustments to the range and levels of micronutrients needed in the population, as dietary patterns and intervention programs evolve.

5. Section 6. If salt is already fortified and has been for decades, how does more fortified salt transform the food system?  Overall, make the link more explicit between fortified salt (and bouillon) and food system transformation.

Response: Salt has been fortified for decades with iodine. However, it’s use as a vehicle for additional micronutrients has been limited. Transformation comes from broadening the range of micronutrients that can be added to salt and salt-containing vehicles. We have made this distinction clearer in the last sentence of section 6 (line 440-441; see highlighted sentence below):

While fortification of salt and salt-containing condiments is not a panacea, it holds enormous potential for being a critical anchor for meeting the dietary requirements across a broad range of micronutrients for those with greatest need, as we collectively move towards a food system that delivers better health for all.

Detailed comments:

6. L49, iron-deficiency anemia. Are the data in reference [4] based on hemoglobin alone or also a measure of iron deficiency.  The author of reference [4] is in the agricultural sector and may not know the importance of distinguishing anemia from iron-deficiency anemia.

Response: The data are based on hemoglobin alone (<12 g/dL).  We have removed the qualifier ‘iron-deficiency’ to reflect anemia alone:

Further, in South Asia, time trend data reveal an increase in anemia among women of reproductive age with increasing production of wheat and rice and lower overall iron-density in the diet [4].      

7. L51-2. Discarding excess cooking water would reduce water-soluble vitamins from unfortified rice.  Are there important vitamins, from a public health perspective, that should be highlighted in this sentence or paragraph?

Response: Losses of B vitamins are also significant, but more so from milling (whole wheat/rice to milled wheat/rice).  The paper capturing mineral losses from excess water was focused on iron and zinc only. We have broadened this sentence to include a broader range of micronutrient losses from a broader range of processing and cooking techniques.

Milling, as well as cooking practices, such as boiling rice in excess water, further reduce the micronutrient content at the point of consumption, particularly for a range of B-vitamins, zinc, and iron [5,75].

8. L80-2. Add a reference(s) for these concepts. 

Response: Given that this text has been revised to state the objective of the paper, we do not believe reference(s) are necessary.

9. How was “healthy” defined in this study?  The same as in studies 13 and 14?

Response: The definition of “healthy” and “unhealthy” has been defined in the footnotes.

10. Please describe what the intended outcome is. 

Response: We have added the qualifier ‘public health’ to describe the intended outcome of interventions to address the consumption of healthier, more micronutrient-dense foods.

An understanding of these geographic and socioeconomic differences in food availability and consumption is necessary to inform the design and targeting of effective policies and interventions that will achieve the intended public health outcomes among the most relevant segments of the population.

11. Sections 2 and 3. The font size (type?) is different between these two sections.  Make uniform across the document. 

Response: We have harmonized the font size throughout the paper. 

12. L120-3. The second part of the sentence refers to fortification, but the first part does not.  Is this a generic statement that is not specific to fortification?

Response: The first part of the statement cites a paper that refers to nutrient-dense foods more broadly, but also specifically lists fortification as one strategy. We further added a caveat about excessive micronutrient or fortification vehicle intake. We have clarified the paragraph as follows:

Improving the nutritional quality of the food supply through increased micronutrient density of foods was identified as a candidate double duty action to address multiple forms of malnutrition [22]. Food fortification is therefore an integral part of food systems change and can be considered a double-duty intervention as long as the program is designed to avoid excessive consumption of micronutrients or of the fortification vehicles themselves.

13. Are availability of and access to health services a further factor?

Response: We are unclear about the sentence to which this question pertains. 

14. Not clear what this means: “have higher health benefit by a consumer encouraging increased consumption”. 

Response: We have updated the text to give an example and make the link between consumer perception and nutrient claims more explicit.  We also added a reference to support the added text:

“Additionally, fortified salt or bouillon may be seen to have greater health benefits by a consumer (e.g. positive consumer bias towards foods with front of pack nutrient claims [75]) encouraging increased consumption and a higher intake of salt”

15. Add reference(s) to support the second part of the sentence.

Response: There is no reference to cite as there are no studies or evidence to corroborate the concern that fortification leads to increased consumption of salt.

16. L223, 234 and elsewhere. Does the use of titration in this sentence imply laboratory measurement of iodine in salt?

Response: Titration refers to balancing/changing the levels of fortification as the intake of salt/bouillon changes in the population. The term titration in the paper has been clarified by the following addition prior to subsequent use in L223 and 234:

“The public health goals of reducing salt intake while fortifying salt to increase micronutrient intake (e.g. through salt iodization) are indeed compatible, given that the concentration of micronutrients can be adjusted as intake of the salt/bouillon is reduced (further referred to in the paper as titration)”

17. Remove “which”. 

Response.  Thank you for this suggestion. We have removed “which” from this sentence sentence: “…where an iron-containing premix which is added to salt…

18. L289-90. The first part of the sentence is missing a word. 

Response. We cannot seem to find the missing word in these lines.

19. L298-9. Lower case vitamin.

Response: Thank you.  We have replaced the capital V with a lower case v.

20. Any household surveys or specifically income and expenditure surveys?

Response: We refer to “household surveys” broadly because various survey types collect information on household food acquisition/purchase that could be used for this purpose (Household Income and Expenditures Surveys, Living Standards Measurement Surveys, Food Consumption Surveys etc.). However, this may be confusing since the term could refer to any survey administered to a household. We have clarified as follows:

Other possibilities include exploring ways in which secondary data sources can be effectively utilized, such as information collected at the retail level, or through national household surveys, such as Household Income and Expenditures Surveys, that collect information on household food acquisition or consumption [71].